# Cardiometabolic Differences in People Living with HIV Receiving Integrase Strand Transfer Inhibitors Compared to Non-nucleoside Reverse Transcriptase Inhibitors: Implications for Current ART Strategies

**DOI:** 10.3390/v16040582

**Published:** 2024-04-10

**Authors:** Wilhelm A. J. W. Vos, Nadira Vadaq, Vasiliki Matzaraki, Twan Otten, Albert L. Groenendijk, Marc J. T. Blaauw, Louise E. van Eekeren, Kees Brinkman, Quirijn de Mast, Niels P. Riksen, Anton F. H. Stalenhoef, Jan van Lunzen, Andre J. A. M. van der Ven, Willem L. Blok, Janneke E. Stalenhoef

**Affiliations:** 1Department of Internal Medicine, Radboud University Medical Center, 6525 GA Nijmegen, The Netherlands; 2Department of Internal Medicine and Infectious Diseases, OLVG, 1091 AC Amsterdam, The Netherlands; 3Department of Medical Microbiology and Infectious Diseases, ErasmusMC, Erasmus University, 3015 CN Rotterdam, The Netherlands; 4Department of Internal Medicine and Infectious Diseases, Elizabeth-Tweesteden Ziekenhuis, 5022 GC Tilburg, The Netherlands

**Keywords:** metabolome, lipoproteome, integrase strand transfer inhibitor, non-nucleoside reverse transcriptase inhibitor, combination antiretroviral therapy, lipids

## Abstract

In people living with HIV (PLHIV), integrase strand transfer inhibitors (INSTIs) are part of the first-line combination antiretroviral therapy (cART), while non-nucleoside reverse transcriptase inhibitor (NNRTI)-based regimens are alternatives. Distinct cART regimens may variably influence the risk for non-AIDS comorbidities. We aimed to compare the metabolome and lipidome of INSTI and NNRTI-based regimens. The 2000HIV study includes asymptomatic PLHIV (n = 1646) on long-term cART, separated into a discovery cohort with 730 INSTI and 617 NNRTI users, and a validation cohort encompassing 209 INSTI and 90 NNRTI users. Baseline plasma samples from INSTI and NNRTI users were compared using mass spectrometry-based untargeted metabolomic (n = 500) analysis. Perturbed metabolic pathways were identified using MetaboAnalyst software. Subsequently, nuclear magnetic resonance spectroscopy was used for targeted lipoprotein and lipid (n = 141) analysis. Metabolome homogeneity was observed between the different types of INSTI and NNRTI. In contrast, higher and lower levels of 59 and 45 metabolites, respectively, were found in the INSTI group compared to NNRTI users, of which 77.9% (81/104) had consistent directionality in the validation cohort. Annotated metabolites belonged mainly to ‘lipid and lipid-like molecules’, ‘organic acids and derivatives’ and ‘organoheterocyclic compounds’. In pathway analysis, perturbed ‘vitamin B1 (thiamin) metabolism’, ‘de novo fatty acid biosynthesis’, ‘bile acid biosynthesis’ and ‘pentose phosphate pathway’ were detected, among others. Lipoprotein and lipid levels in NNRTIs were heterogeneous and could not be compared as a group. INSTIs compared to individual NNRTI types showed that HDL cholesterol was lower in INSTIs compared to nevirapine but higher in INSTIs compared to doravirine. In addition, LDL size was lower in INSTIs and nevirapine compared to doravirine. NNRTIs show more heterogeneous cardiometabolic effects than INSTIs, which hampers the comparison between these two classes of drugs. Targeted lipoproteomic and lipid NMR spectroscopy showed that INSTI use was associated with a more unfavorable lipid profile compared to nevirapine, which was shifted to a more favorable profile for INSTI when substituting nevirapine for doravirine, with evidently higher fold changes. The cardiovascular disease risk profile seems more favorable in INSTIs compared to NNRTIs in untargeted metabolomic analysis using mass-spectrometry.

## 1. Introduction

The risk for non-AIDS comorbidities in people living with HIV (PLHIV) using combination antiretroviral therapy (cART) remains elevated despite viral suppression [1]. Continuous exposure to cART might play a role in this.

First line cART regimens typically include one or two nucleoside/nucleotide reverse transcriptase inhibitors (NRTIs) combined with an integrase strand transfer inhibitor (INSTI) [2]. Alternatively, non-nucleoside reverse transcriptase inhibitors (NNRTIs) are commonly used as a so-called anchor drug. In comparison with NNRTIs, INSTIs have a faster time to viral suppression, better tolerability, and a lower risk to develop HIV resistance over time [3]. This has resulted in INSTIs being the anchor drug in all four CDC-recommended first-line regimens for PLHIV [2]. However, an increasing body of evidence points toward a link between INSTI use and weight gain and metabolic syndrome [3,4,5]. Moreover, some evidence associates initiating an INSTI with increased cardiovascular events, yet only in the first two years [6]. The exact mechanisms behind these associations are not fully understood and thus are the subject of current evaluations [7,8].

Metabolites are small molecules that are important in cellular metabolism and make up a complex network that is collectively known as the metabolome [9]. Classes of metabolites include amino acids, (small) proteins, organic acids, vitamins, fatty acids, lipids, and food or drug breakdown products. Many factors such as genetics, lifestyle, food intake, but also gut microbiome, disease state and medication influence the metabolome. A specific metabolomic profile might therefore even predict disease [10]. HIV and cART are both known to influence the metabolome [11,12,13]. However, despite long-term cART use, the metabolome does not return to a pre-HIV state [14].

We hypothesize that INSTIs affect the metabolome differently than NNRTIs and that these differences might reveal processes related to INSTI-associated diseases. Therefore, we compared the untargeted metabolome of PLHIV using INSTI versus NNRTI-based regimens in a hypothesis free manner to explore where metabolic differences are present. In addition, we evaluated the expression levels of lipoproteins and lipids (n = 141) in INSTI compared to two commonly used NNRTIs to specifically elucidate clinically relevant differences in the lipid spectrum.

## 2. Materials and Methods

### 2.1. Study Population and Data Collection

The 2000HIV study is a prospective longitudinal cohort study that enrolled 1895 asymptomatic PLHIV between October 2019 and October 2021 in the Netherlands. Inclusion criteria were age ≥ 18 years, ≥6 months on cART, most recent HIV viral load < 200 copies/mL and no signs of concurrent infections or pregnancy [15]. The full 2000HIV study consists of a separate discovery and validation cohorts, which were collected at a simultaneous time but at different sites in the Netherlands. The discovery cohort consists of 1559 participants, the validation cohort of 336. The main idea of using two cohorts is to immediately validate any findings in the primary discovery cohort in a secondary validation cohort. If findings are consistently observed over two cohorts, this reduces the chances of incidental findings. The 2000HIV cohort has been extensively described previously [15]. Nevertheless, the sampling process between the two cohorts was the same. Baseline venous 10 mL EDTA blood samples were collected after ≥4 h of fasting, shipped overnight and processed the next morning. Plasma samples were frozen and stored before metabolite and lipid identification. Baseline and HIV specific characteristics were collected from hospital medical files and data available from the Dutch national ATHENA cohort, a long-term cohort collecting anonymous data of HIV patients in the Netherlands [16]. Extensive details of 2000HIV study have been previously published [15]. Participants using immunomodulatory medication (such as oral prednisolone, methotrexate; n = 20) were excluded.

### 2.2. Ethics

The 2000HIV study protocol was approved by the accredited medical research ethics committee Nijmegen (NL68056.091.81). All participants provided written informed consent. Experimental protocols were conducted following the principles of the Declaration of Helsinki.

### 2.3. Untargeted Metabolomics

Untargeted metabolomics was performed on baseline plasma by flow injection electrospray–time-of-flight mass spectrometry to identify metabolites based on the mass-to-charge ratio (ion m/z). Measurements were performed in collaboration with General Metabolics, LLC, and executed at General Metabolics’ labs according to the methodology described previously [17]. Normalization of the samples was performed using a moving median method. Outliers were detected using principal component (PC) analysis and removed if their mean was more than four standard deviations away from PC1 and/or PC2 (n = 2, Appendix A). We used the metabolite annotation from General Metabolics and Human Metabolome Database (HMDB) (https://hmdb.ca/). For downstream data analysis, we selected metabolites that belong to the serum metabolites database (database version 2021-10-24) from the HMDB (n = 500).

### 2.4. Targeted Lipoproteomics

Using baseline plasma, lipoproteins and lipids were measured in a targeted approach using the Nightingale’s (Mannerheimintie 164a, 00300 Helsinki, Finland) Biomarker Analysis Platform. In short, a total of 141 lipoprotein and lipids were measured using nuclear magnetic resonance spectroscopy. Most measured lipoprotein and lipids showed high correlation with other lipoproteins and lipids in this panel. Therefore, we constructed twelve clusters of highly intercorrelated lipoproteins and lipid metabolites (r > 0.75) that encompassed 131 lipoproteins and lipids (see Appendix A for full list) [18]. We used unsupervised hierarchical Ward-linkage clustering based on Spearman correlation coefficients. For each cluster, one lipoprotein or lipid was selected as a cluster representative based on expert opinion. This improved result interpretation and lowered the burden of correction for multiple testing. In addition, nine lipoproteins and lipids that showed no intercorrelation with other lipoproteins or lipids were separately analyzed simultaneously. In total, 21 independent measurements were used for the differential expression analysis.

### 2.5. Statistics

Comparisons in continuous data of baseline characteristics were made using the Student’s T-test or Mann–Whitney U test depending on the data distribution. Non-continuous baseline characteristics were compared using Pearson’s Chi-square test. For baseline characteristics, comparisons with a *p*-value < 0.05 were considered significant. Metabolomic and lipoproteomic measurements were transformed to follow normal distribution using log2 and inverse rank transformation, respectively. In both metabolomics and lipoproteomics data, we performed differential expression (DE) analysis using a linear regression model with sex at birth (sex) and age as covariates. Metabolomics analysis was tested for confounders using principal component analysis (Appendix A). Multiple testing correction was performed using the false discovery rate (FDR) method. FDR *p*-value < 0.05 in the discovery cohort and nominal *p*-value < 0.05 in the validation cohort were considered significant. Metabolic pathway analysis was performed using the web-based platform MetaboAnalyst (https://www.metaboanalyst.ca/) based on the gene set enrichment analysis (GSEA) algorithm [19]. The rationale is that the collective behavior of multiple metabolites in a pathway is less sensitive to random errors introduced through individual peak assignments [19]. We used the default human library (MFN), which consists of a combination of the Kyoto Encyclopedia of Genes and Genomes (KEGG), Biochemical, Genetic and Genomic (BiGG), and Edinburgh Model libraries. Data were analyzed using R studio version 4.2.2 (31 October 2022).

## 3. Results

### 3.1. Participant Selection and Baseline Characteristics

Participants were analyzed using current cART regimes consisting of an NRTI backbone, in combination with either only an INSTI (“INSTI users”) or only an NNRTI (“NNRTI users”) anchor drug (Appendix A). In total, 1646 participants were eligible for analysis. The discovery cohort encompassed 730 INSTI users and 617 NNRTI users, and the validation cohort included 209 INSTI and 90 NNRTI users. Division per INSTI or NNRTI anchor drug can be found in Table 1 and Appendix A. In the discovery cohort, the most common INSTI in use was dolutegravir, and the most common NNRTI in use was nevirapine.

Baseline characteristics are described in Table 1. In the discovery cohort, INSTI users were slightly younger than NNRTI users (51 vs. 53 years, *p* = 0.0001), had known their HIV diagnosis for less time (10.8 vs. 14.6 years, *p* < 0.0001), had been on cART for less time (8.3 vs. 11.5 years, *p* < 0.0001) and had a higher CD4 nadir (280 vs. 240, *p* < 0.0001). Notably, cholesterol-lowering drug use was similar between the two groups.

### 3.2. Untargeted Metabolomic Profiling in INSTI and NNRTI Users

We compared metabolite expression and pathway perturbation between INSTI and NNRTI users. Untargeted metabolomics using mass spectrometry measured 1720 unique metabolites from 1629 participants in both cohorts. To focus on metabolites that play a role in the human metabolic pathways, only metabolites known to be present in serum based on the HMDB database (n = 500) were selected. There were four different types of INSTI and four different types of NNRTI in use in our study (Appendix A). To ensure homogeneity between the different types of INSTI and NNRTI, we performed PC analysis within the INSTI and NNRTI groups itself. Appendix A shows that there was considerable overlap within the first two PCs for both INSTI and NNRTI with respect to metabolite levels. This indicates that the metabolite levels of INSTI and NNRTI users were homogeneous. We therefore analyzed metabolite differences comparing the group of INSTI and NNRTI users. We investigated confounders for metabolites using principal component analysis (Appendix A). Only age had a significant influence on metabolite levels and was subsequently corrected for. In addition, from a biological perspective, sex is an important known factor to drive metabolite levels and was therefore also corrected for.

Next, we performed differential expression analysis of metabolites comparing INSTI to NNRTI users, using sex and age as covariates. The INSTI users from the discovery cohort showed 59 metabolites with significantly higher levels compared to NNRTI users (FDR < 0.05), and 45 metabolites were significantly lower (Figure 1). In the validation cohort, we observed 32 higher metabolite levels in INSTI compared to NNRTI users (*p*-value < 0.05) and 39 lower metabolite levels (Appendix A). Comparison of the discovery and validation cohorts showed 14/59 (23.7%) reproducibly higher metabolites and 24/45 (53.3%) reproducibly lower metabolites (Appendix A). The moderate replication of significant differentially expressed metabolites in the validation cohort is possibly due to power issues since the direction of effect size (increase or decrease) was consistent for 41/59 (69.5%) higher and 40/45 (88.9%) lower metabolites (Appendix A).

Next, we annotated the differentially expressed metabolites from the discovery cohort to metabolite annotations according to General Metabolics annotation and sorted them into metabolic categories based on the HMDB database (Figure 2). As metabolite ion m/z ratios might have several metabolite annotations, one metabolite could match to several metabolite annotations (due to overlapping mass to flight time). The 104 differentially expressed metabolites from the discovery cohort were matched to 205 metabolite annotations. The major categories of differentially expressed annotated metabolites in the discovery cohort were ‘lipid and lipid-like molecules’ (53/205, 25.9%), ‘organic acids and derivatives’ (42/205, 20.5%) and ‘organoheterocyclic compounds’ (32/205, 15.6%). Analysis of the validation cohort validated (*p*-value < 0.05) 24.5% of ‘lipid and lipid-like molecules’ metabolites, 40.5% of ‘organic acids and derivatives’ metabolites and of 75.0% of ‘organoheterocyclic compounds’ metabolites.

Subsequently, we performed pathway analysis using MetaboAnalyst software. In the discovery cohort, significant (*p*-value < 0.05) upregulation of two pathways and downregulation of seven pathways in INSTI users compared to NNRTI users was found (Figure 3). In the validation cohort, we validated ‘caffeine metabolism’, ‘vitamin B1 (thiamin) metabolism’ and ‘keratan sulfate biosynthesis’ (Appendix A). In addition, three significant pathways from the discovery cohort were found in the validation cohort with equal directionality: ‘pentose phosphate pathway’, ‘de novo fatty acid biosynthesis’, and ‘starch and sucrose metabolism’.

Finally, because there were baseline differences between INSTI and NNRTI users, we did a matching of INSTI to NNRTI for important baseline characteristics. Participants were matched on age, sex, BMI, current smoking status, time since HIV diagnosis, time on cART, CD4 nadir, most recent CD4 count, current use of cholesterol lowering medication and medical history of cardiovascular disease. Of the 104 significantly differentially expressed (FDR < 0.05) metabolites in the discovery cohort from the original comparison, 99 (95.2%) were also significantly differentially expressed (*p*-value < 0.05) in the matched discovery cohort. Of the remaining five metabolites, four had a *p*-value < 0.01. This shows that our findings are independent of the matched factors described above.

### 3.3. Targeted Lipoproteomic Profiling in INSTI and NNRTI Users

We observed that ‘lipid and lipid-like molecules’ were one of the main differentially expressed metabolites between INSTI and NNRTI. However, the direct clinical significance of many of these metabolites are unknown. Therefore, we aimed to investigate differences in lipoproteins and lipids between INSTI and NNRTI, focusing thereby on well-known cardiovascular markers. Lipoproteomic measurements were available for 1622 participants using twelve cluster representatives and nine individual and lipoprotein and lipids (Appendix A). Lipoproteins and lipids were analyzed per cluster if they showed high intercorrelation (r > 0.75).

First, we assessed whether there was homogeneity between the different types of INSTI and NNRTI using PC analysis (Appendix A). INSTI showed considerable overlap in PC1 and PC2, whereas the overlap in NNRTI was limited, especially in PC1. This indicates that lipoprotein and lipid levels depend on the type of NNRTI in use. Therefore, we could not compare lipoproteomic levels of INSTI users to the NNRTI user group as a whole. The two most commonly used NNRTIs in the discovery cohort were nevirapine and doravirine (Appendix A), which also showed the least overlap in PC analysis (Appendix A). We therefore compared the findings of INSTI to those of nevirapine and to doravirine separately and, in addition, compared the findings of nevirapine to those of doravirine. Baseline characteristics between these three groups can be found in Appendix A. Differential analysis between groups was performed using a linear model with sex and age as covariates.

First, we compared all INSTI users to nevirapine users in the discovery and validation cohorts (Figure 4A). We found that INSTI users compared to nevirapine users showed a downregulation of two HDL cholesterol clusters and the unsaturated cluster, which is an indicator of the degree of unsaturation in free fatty acids. All these three clusters are negatively associated with cardiovascular disease (CVD) risk. Lower levels of omega-3 should be attributed to differences in food intake. Downregulation of the IDL cholesterol cluster, which is positively and causally associated with CVD, is seen in both cohorts, however, without reaching the threshold of significance in the validation cohort.

Next, we compared all INSTI users to doravirine users in the discovery and validation cohorts (Figure 4B). Here, in contrast, we observed that INSTI users compared to doravirine users showed upregulation of different HDL cholesterol clusters as well as a downregulation of the LDL-size. Smaller LDL-sizes indicate a higher CVD risk. Upregulation of IDL cholesterol cluster and omega-3 is seen in both cohorts; however, significance was only reached in the discovery cohort.

Lastly, we compared nevirapine users to doravirine users in the discovery and validation cohorts (Figure 4C). We found significant upregulation in both cohorts of different HDL clusters, as well as the medium LDL cholesterol cluster in nevirapine users. Smaller LDL-sizes were also observed; although, they were only significantly in the discovery cohort. The fold changes in lipoprotein and lipid levels were much higher when doravirine was part of the analysis, while mostly modest fold changes were noticed when comparing the results in the INSTI and nevirapine group.

## 4. Discussion

In this study, blood metabolic profiles were analyzed in PLHIV using INSTI or NNRTI-based cART regimens to investigate whether systemic metabolomic changes may underly the risk for non-AIDS comorbidities, such as cardiovascular diseases. Using an untargeted metabolome analysis, mainly differences in ‘lipid and lipid-like molecules’, ‘organic acids and derivatives’ and ‘organoheterocyclic compounds’ were found. Pathway analysis showed downregulation of ‘thiamin (vitamin B1) metabolism’, ‘caffeine metabolism’ and ‘keratan sulfate biosynthesis’, and upregulation of ‘bile acid biosynthesis’ both in the discovery and validation cohorts (both *p*-value < 0.05). Additionally, several enriched pathways from the discovery cohort (*p*-value < 0.05) had similar directionality in the validation cohort, including ‘pentose phosphate’ and ‘de novo fatty acid biosynthesis’ pathways. Subsequently, a targeted metabolome analysis was performed, focused on lipoproteins and lipids. Surprisingly and in contrast to INSTI use, lipoprotein and lipid profiles differed between nevirapine and doravirine, the most common NNRTI. We found that nevirapine users showed a more favorable lipoprotein and lipid profile compared to INSTI, while INSTI profile was more favorable compared to doravirine. The fold changes were approximately three times higher in the INSTI vs. doravirine comparison, as opposed to the INSTI vs. nevirapine comparison.

The ‘pentose phosphate pathway’ was downregulated in INSTI users. This pathway is upregulated in untreated HIV and does not return to normal on cART [14,20]. This pathway is important in alternative energy synthesis and production of biomolecules important in HIV infection [14]. HIV infection exploits pathways in T-cells that are normally required for T-cell activation to induce the high energy yield required for HIV proliferation. Normally, oxidative phosphorylation is upregulated in activated T-cells to produce ATP, which also lead to increases in harmful reactive oxygen species. Alternatively, the pentose phosphate pathway is able to use glucose to produce the antioxidant-enhancing NADPH to prevent this damage to cells [21]. HIV-infected cells increase their energy usage to form virions and the upregulated pentose phosphate pathway is thought to induce increased protection on these infected cells. In addition, the pentose phosphate pathway produces deoxynucleotide triphosphates (dNTPs) that play a role in reverse transcriptase activation [21]. Lower levels of dNTPs are associated with impaired HIV transcription [22]. Finding further downregulation in INSTI users might indicate further normalization of the ‘pentose phosphate pathway’ immunometabolism in PLHIV.

The interpretation of the downregulation of thiamin metabolism and keratin sulfate biosynthesis is unclear. An association of INSTI use and thiamin levels has not been described previously. Potentially, this is a consequence of the downregulation of the ‘pentose phosphate pathway’ as thiamin provides a cofactor for transketolase which induces glycolytic intermediates in the ‘pentose phosphate pathway’ [23,24]. Alternatively, HIV has been associated with impaired levels of thiamin producing bacteria in the microbiome, which are only partially recovered during cART [25]. INSTI and NNRTI may differentially influence the restoration of thiamin producing microbiome. Lastly, INSTI could inhibit thiamin metabolism directly through mechanisms yet unknown. Also, keratan sulfate is an glycosaminoglycan that has not been linked to cART, HIV, weight gain, metabolic syndrome or cardiovascular disease [26].

The observation of decreased caffein metabolism in INSTI users is somewhat contradictory. Caffein is mainly metabolized through CYP1A2 [27]. CYP1A2 inhibiting properties have not been reported for INSTIs. In contrast, very moderate CYP1A2 inhibition has been described for NNRTIs efavirenz and nevirapine [28]. In addition, smoking is a well-known CYP1A2 inducer [29]. However, the number of smokers was higher in INSTI than in NNRTI users in the discovery cohort. Therefore, these moderate inhibition and inducing properties are not in line with our observations of reduced caffein metabolism in INSTI users. Possibly, factors outside our considerations influence CYP1A2 metabolism in INSTI compared to NNRTI.

Also, we observed decreased ‘de novo fatty acid biosynthesis’ and increased ‘bile acid biosynthesis’ in INSTI users compared to NNRTI users. ‘De novo fatty acid biosynthesis’ is an endogenous pathway that produces fatty acids primarily from carbohydrates after a high carbohydrate meal, which can be further converted into triglycerides. The clinical consequences of increased ‘de novo fatty acid biosynthesis’ is controversial; although, connections to increased risk for cardiovascular disease and/or diabetes type 2 have been described [30,31]. Decreased ‘de novo fatty acid biosynthesis’ activity could be protective rather than increasing CVD risk. In addition, increased ‘bile acid biosynthesis’ is associated with increased uptake of LDL cholesterol by the liver and conversion into bile acid. Reduced LDL cholesterol in blood lowers the risk for cardiovascular disease.

As both these pathways are lipid related, and ‘lipids and lipid-like molecules’ were one of the main differentially expressed metabolites, we further investigated targeted lipoprotein and lipid changes between INSTI and NNRTI, with focus on known CVD biomarkers. In contrast to the metabolome, we found that lipoprotein and lipid levels differed significantly between the various types of NNRTI. Therefore, we analyzed the difference between INSTI and the two most commonly used NNRTIs, nevirapine and doravirine. These NNRTIs also displayed the most pronounced discrepancy in the PC analysis. INSTI users compared to nevirapine showed a less favorable CVD profile as INSTI users had lower HDL cholesterol clusters with a maximum 0.5 fold change, as well as a higher degree of unsaturated free fatty acids. Indeed, an increase in HDL cholesterol under nevirapine use has been previously described [32]. Whether this also translates into decreased cardiovascular risk for nevirapine users is unclear as large trials are lacking and circumstantial evidence is ambiguous [33,34]. In contrast, INSTI users showed an up to 1.5 fold increase in HDL cholesterol clusters compared to doravirine users, indicating a more favorable CVD lipid risk profile in INSTI users. Furthermore, we observed that the LDL size was smaller in the INSTI compared to the doravirine group which is unfavorable as a smaller LDL size has been associated with an increased cardiovascular risk [35]. We cannot exclude that this effect may come from doravirine, as LDL size was also smaller in nevirapine compared to doravirine users; although, here significance was only observed in the discovery cohort.

Despite our observation in the metabolome of increased ‘bile acid biosynthesis’ and ‘de novo fatty acid biosynthesis’, we did not observe validated changes in LDL or VLDL lipoproteins in INSTI compared to either NNRTI group.

The current study has several strengths and limitations. As far as we know, this is the first time an untargeted metabolome measurement head-to-head comparison has been performed in PLHIV using these different cART regimens. With the analysis of 1646 participants over the two cohorts combined with the 500 metabolites and 141 lipoproteins and lipids, the current study database is extensive. The introduction of a validation cohort allowed to validate findings in an independent cohort. Moreover, we were able to combine the latest state of the art methods in metabolome, lipoprotein and lipid profiling in a well-defined cohort of PLHIV based on an extensive clinical database (ATHENA cohort).

Our study also has limitations. First, we only analyzed 500 metabolites out of the 1720 metabolites measured. As not all plasma metabolites can be annotated adequately with current techniques, we decided to focus on known metabolites. Second, there were considerable differences in the baseline characteristics between the compared groups, which could explain some of the observed differences. However, analysis into confounders revealed no other confounders than we corrected for. Third, our participants were asked to restrain from food intake four hours before sampling. This is somewhat short for lipoproteome and lipid analysis. In addition, our data are only cross-sectional and do not contain any longitudinal data. The stability of our findings over time are therefore uncertain. Also, due to local guidelines the INSTI and NNRTI in use differed between discovery and validation cohorts. As a result, only five participants used doravirine in the validation cohort. Still, we were able to validate several findings. Furthermore, all our findings are correlations and cannot be used to establish causal relations. Finally, we were not able to validate all our findings in the validation cohort. However, a number of reasons might explain this. First, the most significant metabolome and lipoprotein and lipid differences in the discovery cohort were based on small to modest fold changes. However, they did often show the same directionality in the validation cohort, which could indicate insufficient statistical power. Second, there are baseline characteristic differences between the discovery and validation cohorts; this might impede reproducibility in some findings.

## 5. Conclusions

In conclusion, using untargeted metabolomic analysis in INSTI compared to NNRTI users, cardiometabolic risk profile seemed more favorable in INSTI users. However, targeted analysis showed that INSTI use was associated with a more unfavorable lipid profile compared to nevirapine, that was converted into a more favorable profile and three times higher fold change differences substituting nevirapine to doravirine. The more heterogeneous metabolic effects of NNRTI, compared to INSTI, hamper the direct comparison between these two classes of drugs and the formulation of general conclusions.

## Figures and Tables

**Figure 1 viruses-16-00582-f001:**
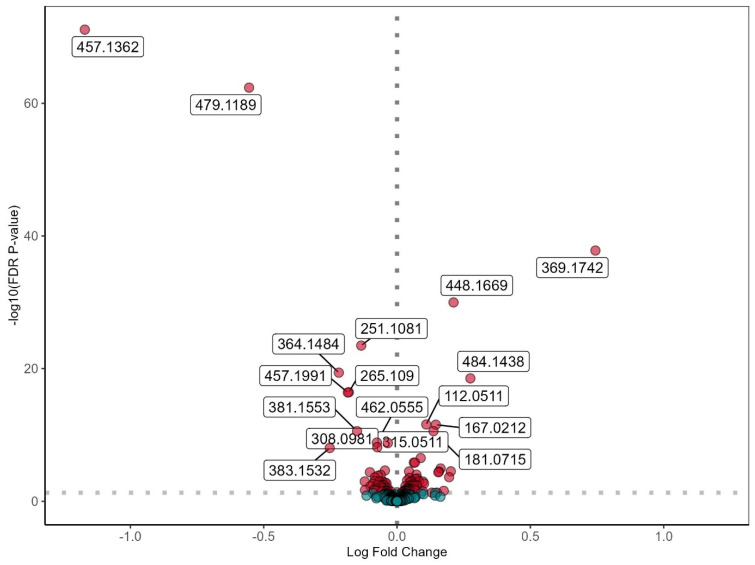
Differentially expressed metabolites in INSTI users compared to NNRTI users in the discovery cohort (INSTI n = 721; NNRTI n = 612). Differential expression analysis using a linear model with sex at birth and age as covariates on 500 metabolites known to be present in serum. *Y*-axis shows the FDR-corrected *p*-value through −log10(FDR), *x*-axis shows the log fold change. Horizontal dotted line represents border of significance (FDR corrected *p*-value < 0.05), vertical dotted line represents border between higher levels (right) and lower levels (left) of metabolites in the INSTI group. Significantly differentially expressed metabolites are shown in red. Non-significant metabolites are shown in green. Numbers specify the significant differentially expressed ion m/z ratio, using untargeted metabolomics. In the discovery cohort, 59 metabolites were significantly higher, and 45 metabolites were significantly lower in INSTI users compared to NNRTI users.

**Figure 2 viruses-16-00582-f002:**
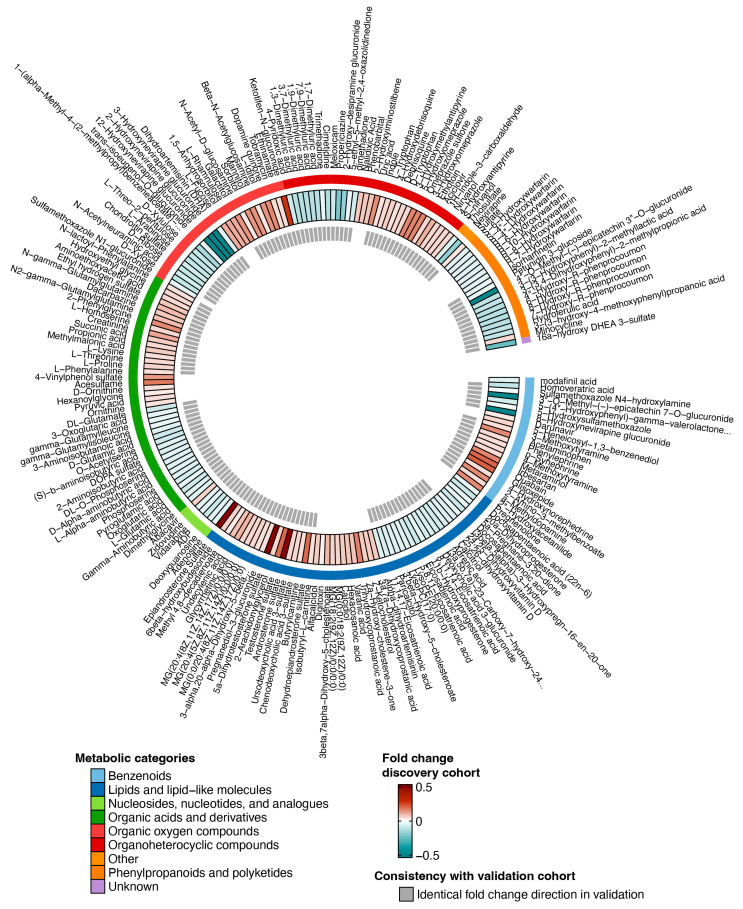
Differentially expressed annotated metabolites in INSTI users compared to NNRTI users in the discovery cohort (INSTI n = 721; NNRTI n = 612) after differential expression analysis on metabolites. Annotated metabolite names are shown on the outside. Three-layer heatmap displays, in order from outside to inside, category of the metabolite, the fold change in the discovery cohort and consistency in directionality in the validation cohort.

**Figure 3 viruses-16-00582-f003:**
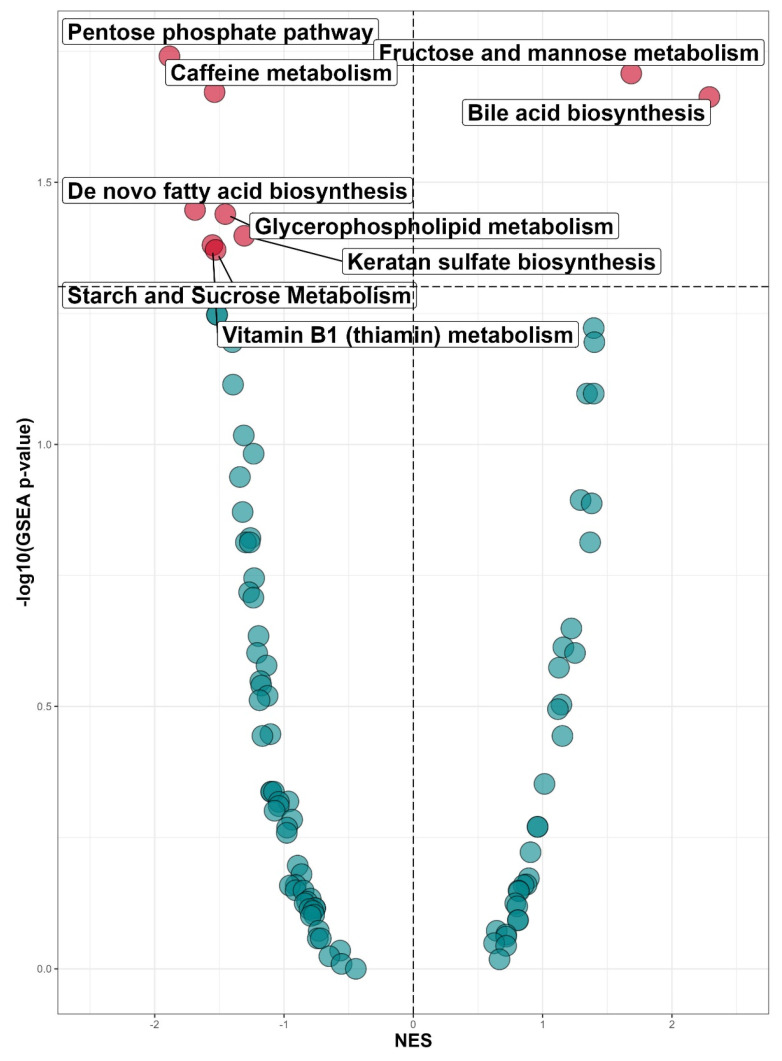
Up- and downregulation of metabolic pathways in INSTI users compared to NNRTI users in the discovery cohort (INSTI n = 721; NNRTI n = 612) using MetaboAnalyst software. *Y*-axis shows *p*-value through −10log(*p*-value), with the horizontal dotted line the threshold for significance (*p* < 0.05). *X*-axis indicates a normalized enrichment score (NES). Positive NES indicates an upregulation, and a negative NES indicates downregulation. Red dots indicate significantly differentially expressed pathways. In the discovery cohort, seven pathways were significantly downregulated, two pathways were upregulated in INSTI users compared to NNRTI users.

**Figure 4 viruses-16-00582-f004:**
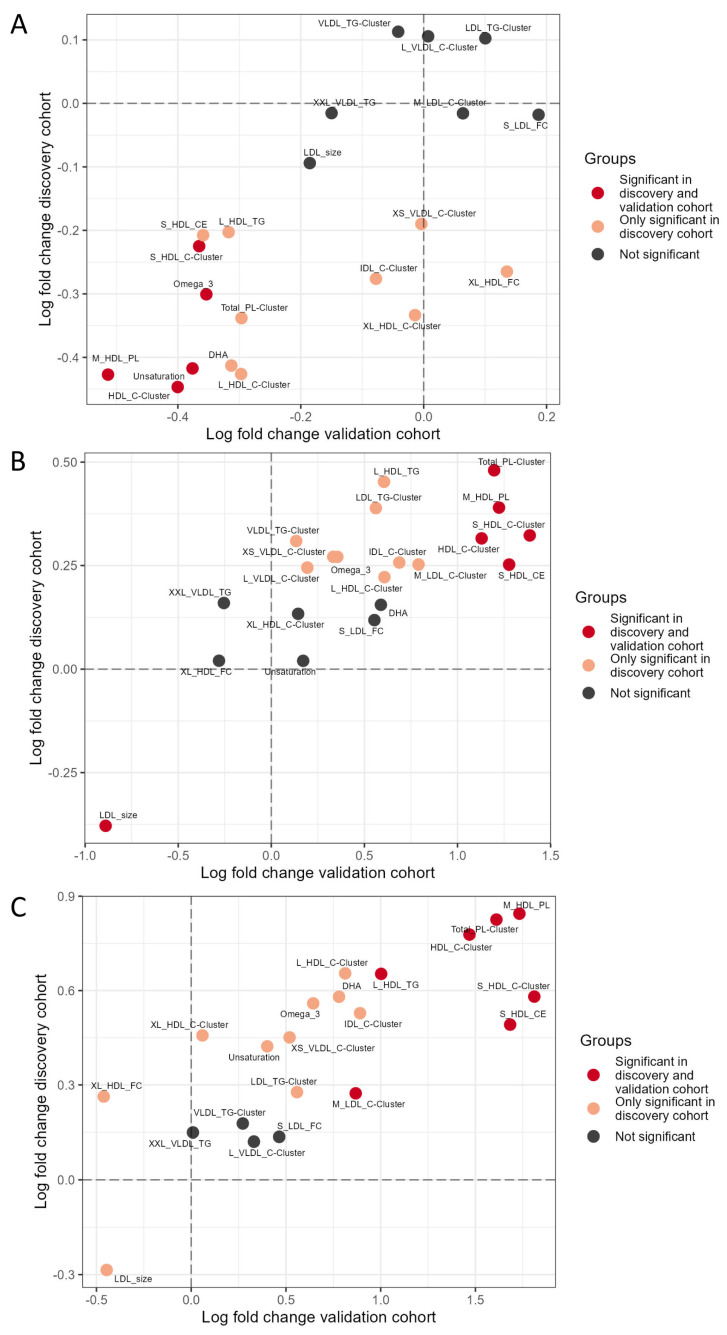
Differential expression of targeted lipoprotein and lipid analysis in INSTI users, NNRTI nevirapine users and NNRTI doravirine users. Differential expression analysis using a linear model with sex at birth and age as covariates on twelve lipoproteins and lipid cluster representatives and nine individual uncorrelated lipoproteins and lipids. Lipoproteins and lipids included per cluster can be found in Appendix A. *Y*-axis demonstrates the log fold change in the discovery cohort, the *X*-axis shows the log fold change in the validation cohort. Dot color indicates whether lipoproteins and lipids were significant differentially expressed in both the discovery and validation cohorts (red), only the discovery cohort (orange) or not significant in the discovery cohort (black). (**A**): Comparison of lipoprotein and lipid levels in INSTI users compared to NNRTI nevirapine users. (**B**): Comparison of lipoprotein and lipid levels in INSTI users compared to NNRTI doravirine users. (**C**): Comparison of lipoprotein and lipid levels in NNRTI nevirapine users compared to NNRTI doravirine users.

**Table 1 viruses-16-00582-t001:** Baseline characteristics of INSTI and NNRTI users in the discovery (INSTI users n = 730, NNRTI users n = 617) and validation cohort (INSTI users n = 209 and NNRTI users n = 90).

	Discovery Cohort			Validation Cohort		
	INSTI	NNRTI	*p*-val	INSTI	NNRTI	*p*-val
	n = 730	n = 617		n = 209	n = 90	
Age in years (IQR)	51.0 (41.0–58.0)	53.0 (46.0–60.0)	0.0001	52.0 (45.0–61.0)	54.0 (48.0–60.0)	0.27
Sex at birth (male)	619 (84.8%)	533 (86.4%)	0.44	178 (85.2%)	74 (82.2%)	0.60
BMI in kg/m^2^ Median (IQR)	25.0 (22.7–27.7)	24.7 (22.2–27.4)	0.061	25.8 (23.0–28.4)	25.1 (22.6–27.4)	0.25
Ethnicity (white)	543 (74.4%)	454 (73.6%)	0.71	180 (86.1%)	78 (86.7%)	1.0
Non-white	185 (25.3%)	163 (26.4%)		29 (13.9%)	12 (13.3%)	
Missing	2 (0.3%)	0 (0.0%)		0 (0.0%)	0 (0.0%)	
HIV duration in years Median (IQR)	10.8 (5.7–16.5)	14.6 (9.9–20.4)	<0.0001	7.0 (3.5–12.7)	14.1 (10.3–19.4)	<0.0001
cART Duration in years Median (IQR)	8.3 (4.7–13.4)	11.5 (8.3–17.9)	<0.0001	6.0 (3.3–10.6)	11.2 (7.8–17.9)	<0.0001
Latest CD4 count (×10^6^ cells/L) Median (IQR)	740.0 (550.5–940.0)	698.5 (550.2–917.5)	0.096	660.0 (480.0–810.0)	685.0 (512.5–867.5)	0.16
CD4 Nadir (×10^6^ cells/L) Median (IQR)	280.0 (160.0–422.5)	240.0 (150.0–350.0)	<0.0001	290.0 (170.0–452.5)	280.0 (145.0–365.0)	0.072
Viral Load Zenith (copies/mL) Median (IQR)	97,000.0 (36,650.5–248,154.0)	100,000.0 (40,000.0–262,000.0)	0.36	156,748.0 (39,240.8–346,862.5)	200,000.0 (70,264.0–387,309.0)	0.20
Currently smoking	238 (32.6%)	167 (27.1%)	0.013	59 (28.2%)	31 (34.4%)	0.40
Missing	56 (7.7%)	35 (5.7%)		23 (11.0%)	7 (7.8%)	
Packyears Median (IQR)	6.0 (0.0–22.0)	4.5 (0.0–21.5)	0.73	6.0 (0.0–28.5)	10.3 (0.0–34.0)	0.33
Had Non-AIDS malignancy	28 (3.8%)	30 (4.9%)	0.42	12 (5.7%)	3 (3.3%)	0.57
Had previous cardiovascular disease	214 (29.3%)	209 (33.9%)	0.077	66 (31.6%)	29 (32.2%)	1.0
On lipid lowering medication	141 (19.3%)	112 (18.2%)	0.62	40 (19.1%)	22 (24.4%)	0.35
INSTI in use:						
Dolutegravir	435 (60%)			77 (37%)		
Bictegravir	164 (22%)			39 (19%)		
Elvitegravir	112 (15%			90 (43%)		
Raltegravir	19 (3%)			3 (1%)		
NNRTI in use:						
Nevirapine		236 (38%)			38 (42%)	
Doravirine		164 (27%)			5 (6%)	
Rilpivirine		133 (22%)			28 (31%)	
Efavirenz		84 (14%)			19 (21%)	

Abbreviations: BMI: body mass index; cART: combination antiretroviral therapy; IQR: inter quartile range; INSTI: integrase strand transfer inhibitor; NNRTI: non-nucleoside reverse transcriptase inhibitor. Cardiovascular disease was considered an official diagnosis of myocardial infarction, stroke, peripheral arterial disease and/or hypertension.

## Data Availability

Data are available through the corresponding author upon reasonable request.

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
