# Peer review of "Cardiometabolic Differences in People Living with HIV Receiving Integrase Strand Transfer Inhibitors Compared to Non-nucleoside Reverse Transcriptase Inhibitors: Implications for Current ART Strategies"

_viruses, 2024, doi:10.3390/v16040582_

Round 1

Reviewer 1 Report

Comments and Suggestions for Authors

This is a very well written, complex study, addressing a very relevant topic, particularly when ART-related metabolic abnormalities are becoming a leading cause of morbidity and mortality in HIV+ patients. I think this study provides innovative information, useful and relevant for clinicians and researchers alike. 

I would appreciate some clarifications regarding some methodological aspects. Firstly, I think some information regarding previous exposure to other antiretrovirals might be useful. It would be important to know if the patients included in the study had been exposed to other ARV before or they have been on the same drug class/drug since starting antiretroviral therapy. Some drugs (i.e. didanosine, stavudine) may cause metabolic impairments even years after discontinuing such drug. Hence, clarifying the therapeutic background of patients might be necessary to avoid biases. Secondly, I find the baseline differences between groups fairly relevant, since a younger, more immunocompetent population might present less metabolic abnormalities. This should be commented. 

I also find relevant the differences in terms of antiretrovirals between the discovery and the validation cohorts. I find relevant the significantly higher proportion of patients on Elvitegravir that, due to the use of boosters, has some unique metabolic repercussions. This applies also to the differences in terms of use of doravirine and efavirenz, which have very different lipidic profiles. Hence, some comments on this aspect could be useful. 

Author Response

We thank reviewer 1 for her/his compliments and interesting suggestions for the manuscript.
We agree with the reviewer that previous ART exposure, especially some nucleoside analogues, is a potential confounder. Therefore we investigated exposure to AZT, DDI and D4T (supplementary figure 4) as confounder through principal component analysis, and found no influence of exposure to these drugs to the metabolite levels.

Second, we also agree with the reviewer that other confounders might influence the metabolomics results. Therefore we also performed principal component analysis to investigate if any other factors might influence the levels of metabolites, and added this in the supplementary material figure 4. This analysis showed that age is the only confounder to influence metabolite levels. In our first manuscript all our analyses were already corrected for age and sex. We decided to correct for sex as men a strongly overrepresented in our study and sex is a very well known factor to influence metabolites. Therefore, we believe that our current corrections are sufficient and that further corrections would rather lead to overcorrection that portray the actual differences between INSTI and NNRTI. Nevertheless, we added a part in limitations that there were considerable differences between the groups.

Finally, we have previously investigated whether or not there were differences between the individual INSTI and NNRTI through principal component analysis. Supplementary figure 6  show that Elvitegravir did not have a different metabolomic profile than the other INSTIs as the principal components of elvitegravir largely overlap with the other INSTI. Therefore we decided to regard the INSTI group as a whole rather than individual.

We hope that with these points above addressed reviewer 1 will be able to endorse publication of our manuscript in Viruses

Reviewer 2 Report

Comments and Suggestions for Authors

This manuscript provides new evidence that cardiometabolic risk profile, as measured by 500 metabolites and 141 lipoproteins and lipids, seems favor persons living with chronic HIV infection who receive INSTI when compared with NNRTI users. One major strength is that several observations are already consistent between a discovery cohort (N = 1,347) and a validation cohort (N = 299). The highlighted results, however, are somewhat biased and should require some re-analysis.

Main concerns

1. The baseline patient information clearly indicates that two comparison groups of interest, i.e., INSTI and NNRTI users, differ in many aspects in terms of demographics, duration of HIV infection, as well as duration of antiretroviral therapy, etc. (Table 1). For fair assessment of potential metabolic phenotypes,  these factors should be kept as covariates for statistical adjustments. Yet data analyses, including linear models, only considered age and sex as potential confounders (e.g., Figure 1).

2. In the abstract and main text, the magnitude of differences between the two patient groups are not described in sufficient detail: the p and FDR values may support statistical significance, but the biological impact of various metabolites must depend on the magnitude of differential expression.

3. The validation cohort is actually different from the discovery cohort in many ways (Table 1), which require some explanation.

4. There is no evidence that the metabolic profiles in each patient group are stable enough to justify a single, cross-sectional sampling. In other words, some longitudinal data must be collected to reveal a potential trajectory for each of the pathway-driven products.

5. BMI is a well-established surrogate indication of metabolic conditions, so it may be helpful to test the relationship between metabolites and BMI at the time of sampling.

Author Response

We thank reviewer 2 for her/his compliments and valuable suggestions.
First, in remarks 1 and 5 the reviewer correctly points out that other confounding factors could influence metabolite levels. To investigate this we performed principal component analysis on a variety of potential confounding variables, including demographics, HIV duration, cART duration and BMI and added this to our manuscript (supplementary figure 4). Our analysis shows that indeed age is an important confounder, which should be corrected for, as we have done in our initial analysis. BMI only effects in PC4 and 5 which accounts for factor 0.06 of 4% and 2.7% of variation, respectively. Therefore BMI correction was not applied. Nevertheless, we believe that the extra analysis of these potentially confounding factors improved the manuscript as a whole and we thank the reviewer.
Second, we agree with remark 2 from the reviewer that magnitude of difference is important in determining the biological impact. Our analysis is however more focused on pathways, as one ionMZ might often be representative of several potential metabolites. We believe that pathways are more representative of processes as they are the result of the combination of several differentially expressed metabolites, and therefore we chose this approach.
Also, we extended our explanation on the discovery and validation cohort in the methods section. Although there are some differences between the cohorts, we believe that this might actually increase the strength of our findings: if findings are consistent in different cohorts, they are more likely to be real differences between the used groups. Still, the fact that some differences are not consistently observed could also be attributed to the differences between the cohorts. Nevertheless, we added this also to the limitations of our study.
Furthermore, we fully agree with the reviewer that our single cross sectional sampling is a limitation. Unfortunately we have no longitudinal data on metabolites. This would be an interesting follow-up research in our group. The point of the reviewer is however a very valid one and we have added this point to the limitations of our study.

We hope that with these points above addressed reviewer 2 will be able to endorse publication of our manuscript in Viruses

Reviewer 3 Report

Comments and Suggestions for Authors

I thank the authors for their article, which is devoted to an untargeted metabolome measurement head-to-head comparison in PLHIV using different cART regimens. The results obtained are of significant scientific interest.

Specific Comments:

1. Section 3.1. “The discovery cohort encompassed 730 INSTI users and 617 NNRTI users, the validation cohort 209 INSTI and 90 NNRTI users.” What is the difference between discovery cohort and validation cohort? It's not clear. Why didn't you use a cohort of healthy donors as a control? And patients not taking cART?

Without these controls, it is impossible to talk about the effect of cART on metabolites circulating in the blood of HIV-infected patients.

2. Section 3.2. Mass spectra should be presented in the supplementary.

3. “Untargeted metabolomics using mass spectrometry measured 1720 unique metabolites from 1629 participants in both cohorts.” Are these metabolites unique to HIV-infected individuals? Or are they also present in healthy donors?

4. In the article, the authors repeatedly refer to pictures in the supplementary, this makes reading the article difficult. Can information from the supplementary be transferred to the main part of the article? For example, “There were four different types of INSTI and four different types of NNRTI in use in our study (supplementary figure 3).” The types of drugs should be listed in the text, and the figure data can be entered into Table 1.

5. Figure 2. It is not clear where the differences lie between the INSTI and NNRTI cohorts.

6. Section 4. The authors discuss the risk of concomitant diseases with HIV infection, including cardiovascular diseases. Is there any literature data on the corresponding metabolites in cardiovascular diseases? Data with references should be provided.

7. “This pathway is important in alternative energy synthesis and production of biomolecules important in HIV infection [14].” This is interesting. Please add more details.

8. “The interpretation of the downregulation of thiamin metabolism and keratin sulfate biosynthesis is unclear.” Please make a guess about their role.

9. The conclusion section should be highlighted separately.

Author Response

We thank reviewer 3 for her/his compliments on the scientific interest of our manuscript and extensive interesting suggestions.

First, we apologize for our somewhat limited explanation on the discovery and validation cohort, as the reviewer points out in remark 1. We extended the explanation of the discovery and validation cohort in the method section our manuscript. Also, we extensively described the cohorts previously [1] We hope this this explanations accommodates any unclarities. Our research focuses on differences within the PLHIV group itself. Therefore, healthy controls were not included in the analysis, also we do not have results to compare. Our main research question was to find metabolites associated with the different ART regimens, and since healthy control generally do not use INSTI and NNRTI ART regimens comparable to PLHIV, this data was not available.

Secondly, we agree with remark 2 that open data sharing promotes transparency and reproducibility in scientific research. However, at this stage of the project, we believe it is prudent to withhold the data from public access for the following reasons:

    -Ongoing Nature of the 2000HIV Project: The 2000HIV project is an ongoing endeavor scheduled for completion in 2026. As such, we are actively engaged in data collection, analysis, and interpretation. With several analyses yet to be conducted, we aim to maximize the utility of the data for our research objectives before making it publicly available.

    -Ensuring Comprehensive Utilization of Data: By withholding the data until all analyses are complete, we can ensure that the findings derived from it are comprehensive and robust. Sharing incomplete or preliminary data prematurely may lead to misinterpretation or incomplete understanding of the project's outcomes.

    -Availability Upon Request to Corresponding Author: While we are unable to provide unrestricted public access to the data at this time, we are committed to facilitating data sharing on a case-by-case basis. Researchers interested in accessing the data may contact the corresponding author directly to request access for specific research purposes.

    -Provision of Summary Statistics: To facilitate downstream data analysis, such as meta-analyses or secondary research, we are prepared to provide summary statistics derived from the 2000HIV project. These summary statistics will offer valuable insights into the dataset's characteristics and trends while preserving the integrity of ongoing analyses.

Furthermore, in response to remark 3: these are all the unique metabolites that were measured in our HIV cohort. We do not know if they are unique to PLHIV or whether they are also present in healthy controls.

Also, in remark 4 the reviewer stresses the importance of adding the different INSTI and NNRTI in table 1 for clarity. We have added these to table 1.

In addition, we believe that the volcano plot in figure 2 (mentioned in remark 5) is a convenient way to show differentially expressed metabolites and the spread of their size change in the discovery cohort. Vertical height indicates significance (the higher, the more significant), and the horizontal placing of metabolites indicates up- or downregulation in INSTI compared to NNRTI: left of the vertical line is downregulation, right of the vertical line is upregulation. The y-as indicates the size of the change (further removed from the horizontal line is higher changer). In the manuscript itself we only show the discovery cohort but the volcano plot of the validation cohort can be found in supplementary table 5. Overlapping significant metabolites between the discovery and validation cohort can be found in supplementary table 2.

Regarding remark 6 we do not believe that a listing of 205 differentially expressed metabolites and their function would increase readability of the manuscript. Instead we have focused on mapping these metabolites to pathways, and focus on pathway changes instead. This gives a better overview of processes that are truly different in INSTI vs NNRTI users.

Furthermore, we agree with remark 7 that the pathway important in alternative energy synthesis is of interest. We have added further details on this pathway in the discussion. We hope this further increases the depth of our findings.

Also, regarding remark 8, we believe that our manuscript already provides a clear analysis of many pathways for which we found concrete evidence or found a logical biological mechanism. We are reluctant to guess or hypothesize but rather maintain the readability of the manuscript as a lot of information on pathways is already described.

Finally, we thank the reviewer for remark 9, we have added this header in the manuscript.

We hope that with these points above addressed reviewer 3 will be able to endorse publication of our manuscript in Viruses

Round 2

Reviewer 3 Report

Comments and Suggestions for Authors

In this study, control groups were not included in the analysis. This is a significant drawback of the experiments performed. Unique metabolites were measured in the HIV cohort, but the authors do not know whether they are unique to PLHIV or whether they are also present in healthy individuals. The manuscript cannot be published without this data.

Also in this manuscript, the Authors consider it prudent to withhold some data from public access until the completion of their HIV 2000 project in 2026. Unfortunately, without this data, the article is not a complete study.